# U.S. CDC support to international *SARS-CoV-2* seroprevalence surveys, May 2020–February 2022

**Amen Ben Hamida**[1,2]*, **Myrna Charles**[2,3], **Christopher Murrill**[2,4], **Olga Henao**[1,2], **Kathleen Gallagher**[1,2]

**1** Division of Global Health Protection, U.S. CDC, Atlanta, GA, United States of America, **2** COVID-19 International Task Force, U.S. CDC, Atlanta, GA, United States of America, **3** Influenza Division, U.S CDC, Atlanta, GA, United States of America, **4** Global Immunization Division, U.S. CDC, Atlanta, GA, United States of America

* ABenHamida@cdc.gov

**Data Availability Statement:** De-identified data uploaded with the submission as supporting materials.

## Abstract

*SARS-CoV-2* seroprevalence surveys provide critical information to assess the burden of COVID-19, describe population immunity, and guide public health strategies. Early in the pandemic, most of these surveys were conducted within high-income countries, leaving significant knowledge gaps in low-and middle-income (LMI) countries. To address this gap, the U.S. Centers for Disease Control and Prevention (CDC) is supporting serosurveys internationally. We conducted a descriptive analysis of international serosurveys supported by CDC during May 12, 2020–February 28, 2022, using an internal tracker including data on the type of assistance provided, study design, population surveyed, laboratory testing performed, and status of implementation. Since the beginning of the pandemic, CDC has supported 72 serosurveys (77 serosurvey rounds) in 35 LMI countries by providing technical assistance (TA) on epidemiologic, statistical, and laboratory methods, financial assistance (FA), or both. Among these serosurvey rounds, the majority (61%) received both TA and FA from CDC, 30% received TA only, 3% received only FA, and 5% were part of informal reviews. Fifty-four percent of these serosurveys target the general population, 13% sample pregnant women, 7% sample healthcare workers, 7% sample other special populations (internally displaced persons, patients, students, and people living with HIV), and 18% assess multiple or other populations. These studies are in different stages of implementation, ranging from protocol development to dissemination of results. They are conducted under the leadership of local governments, who have ownership over the data, in collaboration with international partners. Thirty-four surveys rounds have completed data collection. CDC TA and FA of *SARS-CoV-2* seroprevalence surveys will enhance the knowledge of the COVID-19 pandemic in almost three dozen LMI countries. Support for these surveys should account for current limitations with interpreting results, focusing efforts on prospective cohorts, identifying, and forecasting disease patterns over time, and helping understand antibody kinetics and correlates of protection.

**Funding:** The authors received no specific funding for this work.

**Competing interests:** The authors have declared that no competing interests exist.

## Introduction

Severe acute respiratory syndrome coronavirus 2 (SARS-CoV-2)—the virus responsible for coronavirus disease 2019 (COVID-19)—was first identified in December 2019 in Wuhan, China. After its rapid spread to 20 additional countries in the first six weeks, on January 30, 2020, the World Health Organization (WHO) declared COVID-19 a public health emergency of international concern (PHEIC, [1]). As of May 2022, virtually all countries have been affected by COVID-19, resulting in more than 520 million COVID-19 confirmed cases and more than 6.2 million COVID-19-related deaths reported globally [2].

Seroprevalence surveys are generally conducted to assess the prevalence of pathogen-specific antibodies in the serum and have been used extensively in the past to serve various public health purposes, including to a) assess the prevalence and incidence of infections (e.g., human immunodeficiency virus [HIV], hepatitis C and B viruses) [3–6]; b) identify populations that are susceptible to infections [7]; and c) inform and sustain models for disease control, elimination, or eradication (e.g., measles, rubella, diphtheria, poliomyelitis, and neonatal tetanus) [8–10].

SARS-CoV-2 seroprevalence surveys complement case-surveillance data. Case-surveillance generates data that is needed to assess the burden and severity of disease and identify transmission hotspots. However, these case reports are generally incomplete because many infections are not recognized; for example, if a) an individual is asymptomatic and does not seek health care, b) testing is not available and accessible, or c) identified cased are not appropriately reported to the national public health disease surveillance system. It is in this context that *SARS-CoV-2* seroprevalence surveys—which estimate the prevalence of *SARS-CoV-2* antibodies within a specific population and thus provide a useful proxy to assess previous exposure to and/or vaccination against *SARS-CoV-2*—were initiated. Data from these surveys provide critical information for decision-makers to estimate the true burden of disease, describe population immunity over place and time, identify still-at-risk groups, help assess vaccination coverage, and ultimately guide public health strategies [11].

During the first year of the COVID-19 pandemic, most seroprevalence surveys were conducted within high-income countries, leaving a significant knowledge gap in low- and middle-income (LMI) countries. For example, among the 968 studies included in a systematic literature review of *SARS-CoV-2* seroprevalence studies conducted during 2020, only 23% were from LMI countries [11–13]. To address this gap, the U.S. Centers for Disease Prevention and Control (CDC)—along with multiple other international actors, including WHO—started supporting seroprevalence studies in LMI settings in mid-2020. In this paper, we describe CDC's support of international seroprevalence surveys, define the purpose and characteristics of the supported studies, discuss challenges with implementation and propose possible steps moving forward to use seroprevalence data for public health action.

## Materials and methods

On January 20, 2020, CDC activated its Emergency Operations Center in response to the COVID-19 pandemic to coordinate efforts to stop the transmission of *SARS-CoV-2* [14]. Consequently, an incident management structure was established, including the development and implementation of an incident action plan. Within the Incident Management System, an International Task Force, mostly consisting of CDC staff, was established to oversee CDC's role in the global response, including support for the implementation and monitoring of SARS-CoV-2 international seroprevalence studies. These studies were funded mainly by the Coronavirus Aid, Relief, and Economic Security (CARES) Act, enacted on March 27, 2020 [14–16].

CDC-supported international seroprevalence studies are conducted under the leadership and guidance of local government agencies that request CDC support (e.g., Ministries of Health or national public health institutes) in close coordination with international stakeholders such as the WHO and Africa CDC. The national governmental agencies retain total ownership over generated seroprevalence data. However, limited and temporary access to seroprevalence data may be granted to CDC by the local authorities for a specific task (e.g., data analysis). Technical support is generally provided through CDC offices in country. It includes sharing of templates and guidance documents, assistance in protocol development, advice on sampling methodology and laboratory techniques, staff training, data analysis and interpretation, and dissemination of results.

To ensure some degree of standardization, whenever possible, CDC staff worked closely with WHO, and encouraged countries to use the WHO UNITY protocol entitled "Population-based age-stratified seroepidemiological investigation protocol for COVID-19 infection." The WHO UNITY protocol is part of a global seroepidemiological standardization initiative to increase evidence-based knowledge for action [17]. The WHO UNITY criteria include: using an age-stratified sample from the general population; following a cross-sectional, repeated cross-sectional, or a longitudinal survey design, including all individuals regardless of acute or prior *SARS-CoV-2* infection; and using a validated SARS-CoV-2 laboratory test kit with sensitivity $\geq$ 90% and specificity $\geq$ 97% [17]. Survey designs were modified, as appropriate, to accommodate country-specific needs.

To support the implementation of these surveys, in December 2020 CDC initiated an international seroprevalence community of practice (a virtual monthly group call) to facilitate communication between CDC headquarters and country office staff engaging in seroprevalence work. This community of practice provided an opportunity to discuss and exchange information on topics including sampling, testing, and survey methods and to share ongoing progress, lessons learned, and available seroprevalence data.

In addition, we developed an Excel-based tracker for CDC-supported international seroprevalence surveys, where project staff could update status of protocol development and implementation for each survey. We reviewed information from the tracker and conducted a descriptive analysis of international serosurveys supported by CDC during May 12, 2020–February 28, 2022. Variables included in the analysis were: 1) type of assistance provided (e.g., only technical assistance, only financial assistance, both, informal technical review [as defined by a one-time request for technical review or consultation for a seroprevalence protocol that CDC is not actively engaged in]), 2) WHO region, 3) study design and population surveyed, 4) laboratory testing used, 5) status of survey implementation and 6) status of dissemination.

This activity was reviewed by CDC, deemed not to be research (ID: 0900f3eb81de998d), and was conducted consistent with applicable federal law and CDC policy (See e.g., 45 C.F.R. part 46, 21 C.F.R. part 56; 42 U.S.C. §241(d); 5 U.S.C. §552a; 44 U.S.C. §3501 et seq.). The same applies for all the serosurveys described in this manuscript that have received CDC funding and CDC technical assistance or in which CDC staff are listed as co-authors.

## Results

### Type of CDC support and geographic distribution

As of February 28, 2022, CDC has supported 72 COVID-19 international seroprevalence surveys, corresponding to 77 survey rounds in 35 LMI countries.

Among these survey rounds, the majority (61%) receive both technical and financial assistance from CDC, 30% receive technical assistance only, and 3% receive only financial

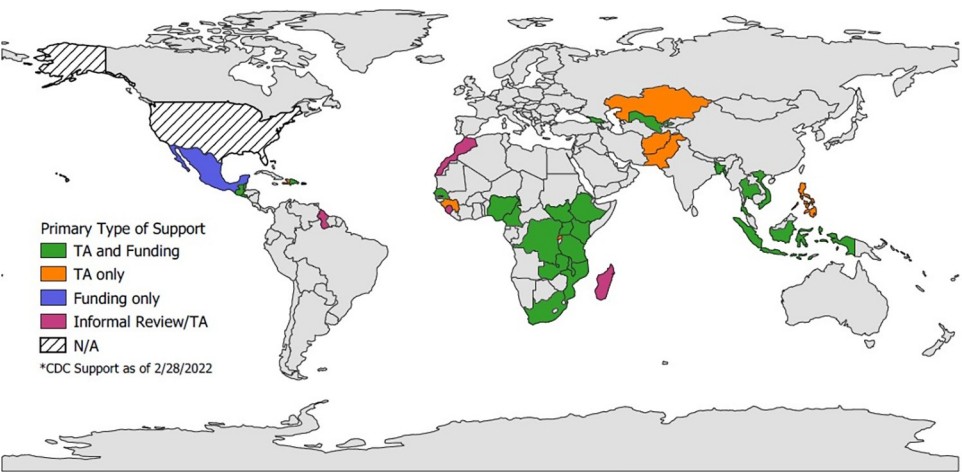

**Fig 1. CDC support to *SARS-CoV-2* international seroprevalence surveys as of February 28, 2022 (N = 72 surveys / 77 survey rounds).** Note: TA: Technical assistance. Greyed countries did not receive CDC assistance. Map developed using QGIS Software and Natural Earth base layer [18,19].

assistance. Six percent were part of informal CDC reviews and technical assistance under the request of local authorities or external partners (Fig 1).

Most surveys (54%) were conducted within the WHO African region, followed by 18% in the Americas region, 13% within the South-East Asian region, 7% in the Western Pacific region, and 4% for each of the European and Eastern Mediterranean regions.

## Study design, population of interest, and WHO UNITY determination

Among the 72 seroprevalence surveys, 69% are based on a cross-sectional design (84% of which are using a single round, while the remaining 16% are using multiple rounds), 19% involve following longitudinal cohorts, and 10% are part of the establishment of COVID-19-specific sentinel surveillance (or being integrated within an already existing surveillance system). The remaining 1% are using a hybrid design customized to the target population (Table 1).

Forty-seven percent of the surveys involve targeting the general population through household-based surveys, and 7% involve targeting the general population through other means (e.g., recruiting from community venues or using a convenience sample); 13% involve sampling pregnant women; 7% involve following health care worker cohorts; and 7% are conducted among other special populations including people living with HIV, persons who were displaced, and students. The remaining 18% target multiple or other populations (e.g., truck drivers).

Sixty-three percent of survey rounds have been assessed against WHO UNITY criteria. Among these, 79% involved methods aligned with the WHO UNITY criteria, and 21% did not.

## Laboratory procedures

Among the 77 serosurvey rounds, 55% involved collecting blood via venipuncture, 25% involved using a finger stick and 5% involved using remnant samples from other studies or ongoing cohorts. Eight percent involved using multiple sample collection types, and 7% are yet to be determined. Seventeen percent involved using dried blood spots to store and transport specimens (Table 1).

**Table 1. Summary of CDC-supported international *SARS-CoV-2* seroprevalence surveys (N = 72 surveys / 77 survey rounds).**

| Variables | N (%) |
|---|---|
| **Type of Support (N = 77 survey rounds)** | |
| Technical and financial assistance | 47 (61%) |
| Technical assistance only | 23 (30%) |
| Financial assistance only | 2 (3%) |
| Informal review | 5 (6%) |
| **Survey design (N = 72 surveys)** | |
| Cross-sectional (one-time survey) | 42 (58%) |
| Repeated cross-sectional | 8 (11%) |
| Longitudinal cohort | 14 (19%) |
| Sentinel surveillance system* | 7 (10%) |
| Hybrid design** | 1 (1%) |
| **Population surveyed (N = 72 surveys)** | |
| General population—Household survey | 34 (47%) |
| General population—Other | 5 (7%) |
| Healthcare workers | 5 (7%) |
| People living with HIV | 3 (4%) |
| Persons who were displaced | 1 (1%) |
| Patients | 1 (1%) |
| Students | 1 (1%) |
| Pregnant | 9 (13%) |
| Other | 3 (4%) |
| Multiple | 10 (14%) |
| **WHO region (N = 72 surveys)** | |
| Africa | 39 (54%) |
| Americas | 13 (18%) |
| South-East Asia | 9 (13%) |
| Europea | 3 (4%) |
| Eastern Mediterranean | 3 (4%) |
| Western Pacific | 5 (7%) |
| **Serologic testing (N = 77 survey rounds)** | |
| ELISA | 40 (52%) |
| Antibody Rapid Test | 8 (10%) |
| Multiplex | 15 (19%) |
| Other/multiple | 9 (12%) |
| To be determined | 5 (6%) |
| **Sample collection (N = 77 survey rounds)** | |
| Venipuncture | 42 (55%) |
| Finger Stick | 19 (25%) |
| Remnant samples | 4 (5%) |
| Multiple | 6 (8%) |
| Not yet determined | 6 (7%) |
| **Progress to date (N = 77 survey rounds)** | |
| Protocol preparation and drafting | 12 (16%) |
| Preparing for data collection | 12 (16%) |
| Data collection ongoing | 13 (17%) |
| Data analysis ongoing | 14 (18%) |
| Drafting of deliverables ongoing | 15 (19%) |
| Dissemination of results complete | 5 (6%) |
| Serosurvey interrupted | 2 (3%) |
| CDC monitoring discontinued | 4 (5%) |

Notes

*Setting new sentinel surveillance systems (e.g., each month, recruiting the first 30 pregnant women who are consulting at an ante-natal care clinic) or integrating seroprevalence within existing ones (e.g., acute febrile illness sentinel surveillance system).

**Customized to the target population. ELISA: enzyme-linked immunosorbent assay. Some variable percentages total 99% due to rounding.

**Table 2. Published CDC-supported *SARS-CoV-2* international seroprevalence studies by February 28, 2022 (N = 5).**

| Country | Geographic Scope | Design | Data Collection Period | Seroprevalence estimate | Infections to Cases Ratio | Reference |
|---------|-----------------|--------|------------------------|-------------------------|---------------------------|-----------|
| **Ethiopia** | 14 major urban areas | Cross-sectional, population-based | June 24–July 8, 2020 | 3.5% (95% CI = 3.2%–3.8%) | 21:1 (in Addis Ababa) | Tadesse et al. [20] |
| **Kenya** | Nairobi | Cross-sectional, population-based | November 2020 | 34.7% (95% CI = 31.8–37.6) | 42:1 | Ngere et al. [21] |
| **Senegal** | National | Cross-sectional, population-based | October 24–November 26, 2020 | 28.4% (95% CI = 26.1–30.8) | 295:1 | Talla et al. [22] |
| **Sierra Leone** | National | Cross-sectional, population-based | March 2021 | 2.6% (95% CI = 1.9%–3.4%) | 43:1 | Barrie et al. [23] |
| **Zambia** | 6 districts | Cross-sectional, population-based | July 4–27, 2020 | Combined PCR and ELISA: 10.6% (95% CI = 7.3–13.9) ELISA: 2.1% (95% CI = 1.1–3.1) | 92:1 | Mulenga et al. [24] |

Notes: Infections to case ratio: Number of possible SARS-CoV-2 infections estimated based on the seroprevalence survey, divided by the number of COVID-19 cases detected by the national surveillance system by the mid-point of the data collection period. ELISA: enzyme-linked immunosorbent assay. PCR: polymerase chain reaction. CI: confidence interval.

The choice of antibody assays has been determined for 95% of serosurvey rounds. About half (55%) involved using an enzyme-linked immunosorbent assay (ELISA) to detect SARS-CoV-2 antibodies in serum, 20% involved using a multiplex assay, and 11% involved using a rapid test. The remaining 13% involved using multiple assays.

## Progress to date

Progress varies across countries. By February 28, 2022, among the 77 survey rounds, 16% are in the stage of protocol development, 16% are in the stage of preparing for data collection, 17% are in the stage of data collection, 18% are in the stage of data analysis, 19% are in the stage of developing their deliverables, and 6% have their dissemination of results completed. The remaining 8% are either interrupted or CDC monitoring was discontinued (e.g., if CDC support was limited to an informal review) (Table 1).

## Dissemination of results

To date, five manuscripts have been published. All five were cross-sectional, population-based surveys conducted in Africa (Table 2). As of February 2022, at least 15 other manuscripts from 10 countries are currently being drafted.

As shown in Table 2, among the published data, SARS-CoV-2 seroprevalence estimates obtained mostly during the first year of pandemic ranged from 2.1% in Zambia (6 districts, July 2020) to 34.7% in Kenya (Nairobi, November 2020) [20–24]. Infection to reported cases ratios—which is the ratio of the number of persons with previous infections as detected by seroprevalence studies (seropositive), to the number of COVID-19 cases detected by the surveillance system during the same period—ranged from 21:1 in Ethiopia (Addis Ababa, June–July 2020) to 295:1 in Senegal (National, October–November 2020) [20–24].

## Discussion

To supplement ongoing COVID-19 surveillance efforts in countries and ensure a thorough understanding of disease prevalence in LMI settings, CDC currently supports more than 70 seroprevalence studies in 35 countries. These surveys vary significantly in type of CDC support

received, study design, the population surveyed, geographic distribution, laboratory method, and progress to date.

The heterogenicity in seroprevalence studies is expected given the varying situations and populations surveyed both within and between countries and has been observed by others conducting a systematic review to track *SARS-CoV-2* serosurveys globally [12,13]. In June 2021, a Canadian team published a systematic literature review of seroprevalence studies conducted during 2020, which included 968 unique serosurveys with significant heterogenicity noted [11]. In order to make sense of this diversity in seroprevalence studies and ensure comparability of results, several global initiatives have been launched including the Serotracker website, designed by the same Canadian team, which conducts a systematic live review of published *SARS-CoV-2* seroprevalence studies, and displays results in an automated live dashboard, allowing users to filter seroprevalence studies by several criteria including region, population of interest, date of data collection, and test type [12,13]. Additional initiatives include the WHO UNITY approach and protocol templates for population-based seroprevalence surveys, which advocate for more standardization and more rigorous methods while conducting serosurveys [17,25]; and the WHO statement on the reporting of seroprevalence studies, which aims to standardize reporting of seroprevalence studies' results [26].

Five CDC-supported seroprevalence studies have been published to date; and at least 15 more had their results disseminated with MOH and are currently in the stage of developing their deliverables. All five published serosurveys were cross-sectional, population-based, and from countries in the WHO Africa region, with wide ranges in both SARS CoV-2 seroprevalence estimates (2.1% to 34.7%) and infection-to-cases ratios (21:1 to 295:1) [20–24]. Of note, the previously mentioned systematic review and meta-analysis, including surveys conducted during 2020, estimated the median corrected SARS CoV-2 seroprevalence to be 0.6% in Southeast Asia, East Asia, and Oceania; 4.1% in high-income countries; 8.2% in North Africa and the Middle East; 10.6% in Latin America and the Caribbean; 12.2% in Central Europe, Eastern Europe, and Central Asia; 17.1% in South Asia; and 19.5% in Sub-Saharan Africa [11]. The extreme variability in infection-to-case ratios from the CDC-supported surveys conducted in 2020 is consistent with those reported in the literature, from a median of 6:1 in Central Europe, Eastern Europe, and Central Asia to 217:1 in Southeast Asia, East Asia, and Oceania [11,13]. This variability highlight the need for health systems building and improved population surveillance in many LMI countries; and could be explained by different factors including differences in public health surveillance systems, availability of testing, clinical presentation, and mitigation measures implemented in-countries.

Seroprevalence results provide critical information for decision makers to guide ongoing public health strategies for the prevention and control of COVID-19. They allow for the assessment of the burden of disease (e.g., cumulative incidence and true case-to-fatality ratio), identification of hidden hotspots, quantification of the infection-to-case ratio, and estimation of vaccination coverage [11,20–24]. When coupled with other surveillance, clinical, and epidemiologic data, data from these surveys can also help distinguish between a) a well-functioning surveillance system capturing most of the cases; b) a real gap in the ability of public health system to identify, diagnose, and report symptomatic cases; and c) a less severe clinical presentation which may lead to limited healthcare-seeking behavior.

Supporting the implementation of international seroprevalence surveys has numerous ongoing and emerging challenges. These include the ever-evolving nature of the pandemic, changes in mitigation strategies and shifting data needs to inform the response, limited resources to support these and other epidemiologic endeavors globally, and competing country priorities. Additionally, legal and logistic challenges limited the capacity of some countries to have access to antibody tests in a timely manner. Finally, there is a clear need for serosurveys

to adopt more efficient processes that allow for timelier implementation. These could include better integration with other ongoing surveillance or epidemiologic efforts; standardization of processes; and real-time analysis and dissemination of results.

Several factors have also challenged the interpretation and usefulness of seroprevalence surveys leading to a change over time in the questions these surveys can address. These include concerns about SARS-CoV-2 cross-reactivity with other pathogens, the implementation of COVID-19 vaccination campaigns, the emergence of new variants, reports of waning immunity, and un-defined correlates of protection (CoP).

Several reports have been shared suggesting possible antigenic cross-reactivity between *SARS-CoV-2* and other pathogens including *Plasmodium* species, Dengue viruses, and other human coronaviruses; possibly leading to a decreased specificity of *SARS-CoV-2* serologic testing and a higher level of false positive results, especially in the African region [27–32]. To help address this, CDC and WHO recommend using serologic tests that have been independently validated and conducting an in-country validation or verification whenever possible [33,34].

Early in the pandemic and before the introduction of *SARS-CoV-2* vaccines, most seroprevalence survey objectives focused on assessing the true burden of disease to help identify gaps in national public health surveillance systems and identify at-risk populations [11,17,25]. The introduction of vaccines, critical in slowing the spread of the virus and decreasing severity of the disease, has created challenges in the design, implementation, and interpretation of *SARS-CoV-2* serosurveys. Testing algorithms involving the use of antibody assays with different targets (e.g. antibodies anti-nucleocapsid protein and antibodies anti-Spike protein) have been developed to help distinguish between natural and vaccine-induced immunity for vaccines that only target Spike protein-derived antigens (e.g., current mRNA and viral vector vaccines) [35]. However, if the countries are using other vaccines or the survey design does not allow to distinguish between natural and vaccine-induced immunity, the objectives of seroprevalence surveys would be mostly limited to elucidating potentially at-risk populations and allocating scarce resources (vaccines) efficiently [35]. Additionally, mounting evidence of waning antibodies with both natural and vaccine-induced immunities challenges the ability to assess both the true burden of disease and vaccination coverage [36–39]. Of note, infection severity, timing of testing, and the choice of antibody assays all seem to significantly impact antibodies' detection over time [40]. CDC is currently working with international partners to address the statistical, epidemiologic, and laboratory considerations of vaccine introduction and to adapt seroprevalence protocols accordingly, based on WHO interim guidance and in-country specific needs and priorities.

Finally, the question of correlates of protection (CoP) for SARS-CoV2 is not resolved. Several studies have documented an inverse relationship between neutralizing antibody levels and incidence of infection, and a correlation between these antibody levels and vaccine efficacy suggests a possible relative CoP against infection that is likely defined by quantitative thresholds [35,41–44]. However, most serosurveys underway are using qualitative assays to assess circulating antibodies. Cellular immunity, which is generally more durable and likely to be integral to CoP, is generally not assessed in these surveys due to technical and cost challenges [35,41].

CDC technical assistance and financial support of SARS-CoV-2 international seroprevalence surveys contribute to strengthening local capacity to conduct epidemiologic surveillance and enhance the knowledge of the COVID-19 pandemic in almost three dozen LMI countries. Considering constantly evolving challenges to implementation and interpretation of seroprevalence results two years into the COVID-19 pandemic, there is a need for the global public health community to assess the best strategies for future SARS-CoV-2 seroprevalence efforts. These strategies could include developing a better understanding of antibody kinetics,

identifying and forecasting epidemic patterns over time, and helping assess CoP. These objectives may be attainable by relying on prospective cohorts, integrating COVID-19 sero-surveillance with surveillance of other pathogens, and introducing quantitative testing of binding and neutralizing antibodies. Support for international seroprevalence surveys can help understand patterns of transmission, guide public health interventions, allocate resources strategically, and ultimately control the spread of the pandemic.

## Supporting information

**S1 Data. De-identified data set.**
(XLSX)

## Acknowledgments

U.S. CDC Countries offices involved in seroprevalence work; National Ministries of Public Health, CDCs, Institutes for Public health, and other implementing partners; U.S. CDC Division of Global Health Protection, Global Epidemiology, Laboratory, and Surveillance Branch, Surveillance and Information Systems Team and Laboratory Team; U.S. CDC International Task Force, Epidemiology and Laboratory Teams; WHO Seroprevalence Team; SeroTracker team; and Africa CDC Seroprevalence Team.

## Disclaimer

The findings and conclusions in this study are those of the author(s) and do not necessarily represent the views of the U.S. Centers for Disease Control and Prevention/Agency for Toxic Substances and Disease Registry.

## Author Contributions

**Conceptualization:** Amen Ben Hamida, Myrna Charles, Christopher Murrill, Olga Henao, Kathleen Gallagher.

**Data curation:** Amen Ben Hamida, Myrna Charles, Kathleen Gallagher.

**Formal analysis:** Amen Ben Hamida.

**Methodology:** Amen Ben Hamida, Myrna Charles, Christopher Murrill, Olga Henao, Kathleen Gallagher.

**Project administration:** Amen Ben Hamida, Myrna Charles, Christopher Murrill, Olga Henao, Kathleen Gallagher.

**Resources:** Christopher Murrill, Olga Henao, Kathleen Gallagher.

**Software:** Amen Ben Hamida, Myrna Charles.

**Supervision:** Christopher Murrill, Olga Henao, Kathleen Gallagher.

**Visualization:** Amen Ben Hamida.

**Writing – original draft:** Amen Ben Hamida.

**Writing – review & editing:** Amen Ben Hamida, Myrna Charles, Christopher Murrill, Olga Henao, Kathleen Gallagher.

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
