## [Decision Letter · Decision Letter 0]

14 Jun 2022

PGPH-D-22-00877

U.S. CDC support to international SARS-CoV-2 seroprevalence surveys, May 2020–February 2022

Dear Dr. Ben Hamida,

Thank you for submitting your manuscript to PLOS Global Public Health. After careful consideration, we feel that it has merit but does not fully meet PLOS Global Public Health’s publication criteria as it currently stands. Therefore, we invite you to submit a revised version of the manuscript that addresses the points raised during the review process.

Please submit your revised manuscript by . If you will need more time than this to complete your revisions, please reply to this message or contact the journal office at globalpubhealth@plos.org. Please include the following items when submitting your revised manuscript:

We look forward to receiving your revised manuscript.

Kind regards,

Julio Croda, Ph.D, M.D.

Academic Editor

Journal Requirements:

1. Please update the 'Competing Interests' statement with this "The authors have declared that no competing interests exist".

a. Please clarify all sources of funding (financial or material support) for your study. List the grants (with grant number) or organizations (with url) that supported your study, including funding received from your institution. 

b. State the initials, alongside each funding source, of each author to receive each grant.

c. State what role the funders took in the study. If the funders had no role in your study, please state: “The funders had no role in study design, data collection and analysis, decision to publish, or preparation of the manuscript.”

d. If any authors received a salary from any of your funders, please state which authors and which funders.

If you did not receive any funding for this study, please simply state: “The authors received no specific funding for this work."

3. In the online submission form, you indicated that “Data shared with reviewers. Not to be published publicly. The authors can consider sharing data upon request.”. All PLOS journals now require all data underlying the findings described in their manuscript to be freely available to other researchers, either 1. In a public repository, 2. Within the manuscript itself, or 3. Uploaded as supplementary information.

Additional Editor Comments (if provided):

Dear Dr Amen Ben Hamida,

Thank you again for submitting your manuscript "U.S. CDC support to international SARS-CoV-2 seroprevalence surveys, May 2020–February 2022" to Plos Global Public Health. We have now received reports from 3 reviewers and, on the basis of their comments, we have decided to invite a revision of your work for further consideration in our journal. Your revision should address all the points raised by our reviewers (see their reports below).

When resubmitting, you must provide a point-by-point response to the reviewers’ comments. Please show all changes in the manuscript text file with track changes or colour highlighting. If you are unable to address specific reviewer requests or find any points invalid, please explain why in the point-by-point response.

Reviewers' comments:

Reviewer's Responses to Questions

**Comments to the Author**

1. Does this manuscript meet PLOS Global Public Health’s publication criteria? Is the manuscript technically sound, and do the data support the conclusions? The manuscript must describe methodologically and ethically rigorous research with conclusions that are appropriately drawn based on the data presented.

Reviewer #1: Partly

Reviewer #2: Yes

Reviewer #3: Yes

2. Has the statistical analysis been performed appropriately and rigorously?

Reviewer #1: Yes

Reviewer #2: N/A

Reviewer #3: Yes

3. Have the authors made all data underlying the findings in their manuscript fully available (please refer to the Data Availability Statement at the start of the manuscript PDF file)?

Reviewer #1: Yes

Reviewer #2: Yes

Reviewer #3: Yes

4. Is the manuscript presented in an intelligible fashion and written in standard English?

Reviewer #1: Yes

Reviewer #2: Yes

Reviewer #3: Yes

5. Review Comments to the Author

Reviewer #1: The manuscript is technically good. It presents all the mandatory sections for an article and follows the steps of a scientific research. However, the results are interesting for the study funder itself and do not impact knowledge about the object of the study for people outside the organization.

Despite being technically flawless, the manuscript is more of a technical report than original research.

Reviewer #2: the article is very interesting. clearly summarizes the CDC's investment in research on the seroprevalence of covid-19. I missed work in Brazil, since it is one of the countries with the highest number of cases and deaths. But this does not rule out the importance of publication.

Reviewer #3: This paper summarized the CDC’s support of 72 international SARS-CoV-2 seroprevalence studies in 35 low and middle income countries. The paper describes the type of support the CDC provided (technical or financial assistance), the geographic distribution of support, study design including laboratory procedures, and population surveyed. The paper was clearly written and describes important ongoing global public health surveillance efforts. The discussion clearly describes the limitations to seroprevalence surveys, including the need for study designs to differentiate between vaccine-induced and naturally-acquired immunity. The manuscript was very clear in its description of CDC activities to date. A few comments:

• L203. The paper mentions that results from 5 serosurveys supported by the CDC have been published to date and 15 more have been shared with MOH. Please clarify the plans for dissemination of the remaining studies.

• Addressing the lack of global access to antibody tests is an urgent global health priority. I would suggest adding a paragraph in the Discussion to discuss access to antibody testing and the glaring need for international support, especially from the US, to improve global equity in access to antibody tests critical for global health surveillance.

• Additionally, I would suggest discussing the glaring disparities in infection-to-case ratios (Table 2) through a lens of health equity. The high infection-to-case ratios identified in many low and middle income countries highlights the critical need for health systems building and improved population surveillance.

6. PLOS authors have the option to publish the peer review history of their article (what does this mean?). If published, this will include your full peer review and any attached files.

**Do you want your identity to be public for this peer review?** For information about this choice, including consent withdrawal, please see our Privacy Policy.

Reviewer #1: No

Reviewer #2: **Yes: **LUCIANO PAMPLONA DE GOES CAVALCANTI

Reviewer #3: No

---

## [Editor Report · Decision Letter 1]

2 Jul 2022

PGPH-D-22-00877R1

U.S. CDC support to international SARS-CoV-2 seroprevalence surveys, May 2020–February 2022

Dear Dr. Ben Hamida,

Thank you for submitting your manuscript to PLOS Global Public Health. After careful consideration, we feel that it has merit but does not fully meet PLOS Global Public Health’s publication criteria as it currently stands. Therefore, we invite you to submit a revised version of the manuscript that addresses the points raised during the review process.

We look forward to receiving your revised manuscript.

Kind regards,

Julio Croda, Ph.D, M.D.

Academic Editor

Journal Requirements:

Additional Editor Comments (if provided):

Please answer reviewers' questions
---

## [Editor Report · Decision Letter 2]

18 Jul 2022

U.S. CDC support to international SARS-CoV-2 seroprevalence surveys, May 2020–February 2022

PGPH-D-22-00877R2

Dear Dr Ben Hamida,

We are pleased to inform you that your manuscript 'U.S. CDC support to international SARS-CoV-2 seroprevalence surveys, May 2020–February 2022' has been provisionally accepted for publication in PLOS Global Public Health.

Best regards,

Julio Croda, Ph.D, M.D.

Academic Editor